# The Effect of Harvesting on National Forest Carbon Sinks up to 2050 Simulated by the CBM-CFS3 Model: A Case Study from Slovenia

**Jernej Jevšenak [1],\* , Matija Klopčič [2] and Boštjan Mali [1]**

1   Slovenian Forestry Institute, Večna pot 2, 1000 Ljubljana, Slovenia; bostjan.mali@gozdis.si
2   Department of Forestry and Renewable Forest Resources, University of Ljubljana, Večna pot 83, 1000 Ljubljana, Slovenia; matija.klopcic@bf.uni-lj.si
\*   Correspondence: jernej.jevsenak@gozdis.si

**Abstract:** With the advent of global warming, forests are becoming an increasingly important carbon sink that can mitigate the negative effects of climate change. An understanding of the carbon dynamics of forests is, therefore, crucial to implement appropriate forest management strategies and to meet the expectations of the Paris Agreement with respect to international reporting schemes. One of the most frequently used models for simulating the dynamics of carbon stocks in forests is the Carbon Budget Model of the Canadian Forest Sector (CBM-CFS3). We applied this model in our study to evaluate the effects of harvesting on the carbon sink dynamics in Slovenian forests. Five harvesting scenarios were defined: (1) business as usual (BAU), (2) harvesting in line with current forest management plans (PLAN), (3) more frequent natural hazards (HAZ), (4) high harvest (HH) and (5) low harvest (LH). The simulated forest carbon dynamics revealed important differences between the harvesting scenarios. Relative to the base year of 2014, by 2050 the carbon stock in above-ground biomass is projected to increase by 28.4% (LH), 19% (BAU), 10% (PLAN), 6.5% (HAZ) and 1.2% (HH). Slovenian forests can be expected to be a carbon sink until harvesting exceeds approximately 9 million $m^3$ annually, which is close to the calculated total annual volume increase. Our results are also important in terms of Forest Reference Levels (FRL), which will take place in European Union (EU) member states in the period 2021–2025. For Slovenia, the FRL was set to $-3270.2$ Gg $CO_2$ eq/year, meaning that the total timber harvested should not exceed 6 million $m^3$ annually.

**Keywords:** carbon; forest reference levels; forest management; harvesting scenarios; yield curves; UNFCCC

## 1. Introduction

Global forests represent one of the most important terrestrial carbon stocks [1] and play an important role in global carbon cycles, particularly due to their carbon sequestration capacity and positive influence on water balance and temperature regulation [2]. Global carbon stocks in forests are estimated at 861 ± 66 Pg C, of which 44% represents forest soils, 42% forest biomass, 8% dead wood biomass and 5% litterfall. Globally, over a half of the carbon stocks in forests can be found in tropical forests (55%), followed by boreal (32%) and temperate forests (14%) [3].

Greenhouse gas emissions and removals by forests are reported annually by countries committed to the United Nations Framework Convention on Climate Change (UNFCCC) [4], in the framework of the land use, land-use change and forestry (LULUCF) sector. This is one of five economic sectors, with the other four being energy, industrial processes and product use, agriculture and waste. The European Union (EU) included the LULUCF sector in its energy and climate policy for the reduction of emissions

by 2030 and attainment of long-term environmental objectives of the Paris Agreement [5]. Beforehand, the European Commission estimated that current policies would not be sufficient to reach the goal of a 30% emissions reduction by 2030 in the sectors that were not included in the EU Emissions Trading Scheme [6]. This made the accounting of emissions and removals, including those from managed forest land, mandatory for all EU member states in the period 2021–2030 [7].

Forest management decisions, such as harvesting regime and intensity, conversion of vulnerable forest types, forest regeneration and promotion of selected tree species, have an important effect on carbon stocks in forests [8]. The change in carbon stocks in forests depends on forest growth, mortality and harvesting, with harvesting being the most important measure when it comes to directing forest development. Slovenia is one of the most forested countries in Europe, and in the period 1991–2013, forestry in Slovenia was characterised by a relatively low total felling, which ranged from 2 to 4 million m$^3$ annually [9]. The reasons for low harvesting in that time were mainly related to socioeconomic changes after Slovenian independence, denationalization and a forestry policy that pursued the goal of accumulating forest growing stocks [10]. At the beginning of 2014, Slovenian forests were damaged by a large-scale ice storm [11], followed by an extensive windstorm in December 2017. In the meantime, mixed spruce forests at secondary sites were additionally affected by bark beetles [12]. Between 2014 and 2018, total felling increased and averaged 5.9 million m$^3$ annually. However, this was still below the total annual yield [9]. In addition to the amount of felling, the ageing of forests is frequently mentioned in connection with Slovenian forests [13,14]. This process is associated with reduced resistance to natural disturbances, i.e., windthrows and bark beetle attacks [15], as well as a reduction in the economic value of forests [16], although positive aspects of ageing are often reported [17,18]. The demand for timber is expected to grow in the future [19,20], which is why understanding the impact of harvesting on the age structure, as well as on the long-term provision of a forest carbon sink, is of key importance.

Decisions related to forest policies and forest management are often based on projections of various forest development models [21,22]. One of the most commonly used models to explore future forest and land-use policy options is the Carbon Budget Model of the Canadian Forest Sector (CBM-CFS3) [23,24] developed by the Canadian Forest Service. The CBM-CFS3 simulates the dynamics of forest carbon pools, taking into account various assumptions such as the forest management method, land-use change, occurrence of natural disturbances and harvesting. The CBM-CFS3 has been shown to be a reliable tool for simulating carbon dynamics in several countries in the Northern Hemisphere, e.g., Canada [25,26], South Korea [27], Russia [28] and Italy [29]. In addition, the CBM-CFS3 has also been used to prepare simulations of forest carbon dynamics for most of the EU member states [30].

Similar simulations have not yet been prepared for Slovenia, but they would be very useful to evaluate the impact of forest management on carbon sink dynamics and provide guidance in establishing future forest policy. In this study, we present simulations of the forest carbon sink with the CBM-CFS3 for the 2014–2050 period. The main objective of the study was to evaluate the effect of harvesting impacts in Slovenia on forest carbon storage according to five harvesting scenarios. The simulations were performed for all forest carbon pools (see Section 2.1), where the focus of our study was on above- and below-ground biomass.

## 2. Materials and Methods

### 2.1. The Carbon Budget Model of the Canadian Forest Sector (CBM-CFS3)

The CBM-CFS3 [23,24] is a freely available model enabling the simulation of changes in carbon stocks in forests using five forest carbon pools: (1) above-ground biomass, (2) below-ground biomass, (3) deadwood, (4) mineral soils and (5) litter. The model operates on the basis of a forest inventory database and yield curves, which describe the development of growing stock in relation to the age of forest stands. The model is in line with the concept of Intergovernmental Panel on Climate Change (IPCC) reporting standards. The spatial scale of operation can range from individual forest stands to

forest types and landscape spatial entities. The simulation results are provided on an annual basis, separately for all five carbon pools [23]. Advanced options for displaying the results enable analyses of the carbon transitions between forest carbon pools, the atmosphere and harvested wood products.

The structure of the CBM-CFS3 model includes three main steps [24]: (1) a pre-processor program which prepares the inventory database and generates the pool for dead biomass, (2) a pre-processor program which calculates the carbon stock for individual pools and localities on an annual basis during the simulation, and (3) an archive index database which includes model parameters and connects them to the input data and simulation results. The changes in below-ground biomass are calculated using the methodology presented by Li, et al. [31]. The archive index database was customised for the EU member states and includes climate information which affects the annual decay rate of dead organic matter, the ecological parameters for individual bio-geographical regions of the EU and the set of volume-to-biomass conversion coefficients for European tree species which are important for the correct conversion of merchantable volume into biomass and foliage components.

The model input data are represented by seven independent matrices. The Classifiers and Values matrix defines the number of model units and the classifiers which define them. In our case, the classifiers were defined by individual forest types and further divided by the share of conifers/broadleaves. The Age Classes matrix defines the range of existing age classes and the degree of transition from one age class to another. The initial state of forests is described in the Inventory matrix, which includes the areas of individual types of forests, further divided into mixture and age classes. The development of woody biomass in forests by age for individual types of forests is determined in yield curves included in the Growth and Yield matrix. Growing stock is defined as the gross merchantable volume of biomass, which includes the volume inside the bark of the main stem, excluding tops and stumps, but including defective and decayed wood of trees or stands [32]. With the Transition Rules matrix, we can define the rules on how individual forest types transit from one to another. For example, a large-area disruption which affects an adult spruce stand can transition to another forest type with a higher share of broadleaves and thus more resilient to natural hazards. All natural and man-made disturbances, including harvesting, are defined in the Disturbance Events matrix. Disturbances can be expressed in absolute spatial units or as an area proportion on which the disturbance can occur. The Disturbance Types matrix precisely defines disturbances and their influence, and later connects them to the disturbance parameters set out in the archive index database.

The inventory database, yield curves, disturbance events and harvest schedule are key data inputs for the simulation in the CBM-CFS3. For each year of simulation, disturbance must be related to a specific forest type and its age class and connected to one of the available disturbance types from the archive index database. There are more than 100 disturbance types available, ranging from forest fires to bark beetle outbreaks and different types of harvesting. In addition, the user also has the possibility of determining their own type of disturbance and the intensity of its influence on forest carbon pools.

The CBM-CFS3 was first developed for use in even-aged forests where the age of trees is known. However, when yield curves realistically describe the development of individual stands of different species, the model can also be used in mixed uneven-aged forests which are typical for Slovenia [32]. Two basic data sources were used to run the model, both provided by the Slovenia Forest Service (SFS): (1) the Forest Compartment Database from 2014 and (2) the Permanent Sample Plot (PSP) Database. The Forest Compartment Database consists of approximately 53,000 forest compartments with a mean size of 22 ha, which are permanent forest planning categories and cover the information on all forests in the country. For each compartment, various forest attributes are available, such as forest area, forest type, growing stock and tree species composition. Using data on forest type, similar forest compartments were aggregated and represented the initial state of the forests in 2014. The PSPs are part of the control sampling method in Slovenia [33]. Each plot (500 $m^2$ each) is remeasured every 10 years, and common tree attributes are surveyed for each tree on the plot, such as location, tree species, DBH, height of selected trees and status between consecutive inventories (e.g., unchanged, harvested, died, ingrowth).

## 2.2. Data Preparation

### 2.2.1. Preparation of Inventory Data

In our study, the model classifiers were defined as a combination of forest type and tree species mixture. Forest type was determined according to the typology of Slovenian forest sites [34], see Table 1. Tree species mixture was represented by three categories based on the prevailing tree species: (1) if the total growing stock of broadleaves within a compartment was greater than or equal to 75%, the forest compartment was classified as broadleaved forests; (2) if the total growing stock of conifers was greater than or equal to 75%, the forest compartment was classified as coniferous forests; and (3) in all other cases, the forest compartment was classified as mixed forests, which was the prevailing category in all forest types (Table 1). Finally, 43 actual model classifiers were defined.

**Table 1.** Forest types defined by Kutnar, Veselič, Dakskobler and Robič [34], assigned climate unit from the archive index database, total share of areas among different forest types and the share of broadleaved (BRD), coniferous (CON) and mixed (MIX) stands within each forest type.

| Short | Forest Type | Climate Unit | Total Share | BRD | CON | MIX |
|-------|-------------|--------------|-------------|-----|-----|-----|
| FT1 | Forests of *Salix* spp. with *Populus* spp., forests of *Alnus glutinosa* and of *A. incana* | Slovenia CLU35 | 0.02 | 0.04 | 0.01 | 0.95 |
| FT2 | Forests of *Carpinus betulus* and of *Quercus petraea* on carbonate and mixed bedrock | Slovenia CLU45 | 0.07 | <0.01 | <0.01 | 0.99 |
| FT3 | Forests of *Carpinus betulus* with *Quercus petraea* on silicate bedrock | Slovenia CLU45 | <0.01 | <0.01 | <0.01 | 0.99 |
| FT4 | Submontane *Fagus sylvatica* forests on carbonate and mixed bedrock | Slovenia CLU55 | 0.16 | <0.01 | <0.01 | 0.99 |
| FT5 | Submontane *Fagus sylvatica* forests on silicate bedrock | Slovenia CLU45 | 0.16 | <0.01 | <0.01 | 0.99 |
| FT6 | Montane, altimontane and subalpine *Fagus sylvatica* forests on carbonate and mixed bedrock | Slovenia CLU55 | 0.13 | <0.01 | 0.01 | 0.99 |
| FT7 | Montane and altimontane *Fagus sylvatica* forests on silicate bedrock | Slovenia CLU54 | 0.08 | 0.01 | 0.02 | 0.97 |
| FT8 | Forests of *Fagus sylvatica* with *Abies alba* | Slovenia CLU55 | 0.14 | 0.01 | 0.01 | 0.98 |
| FT9 | Forests of *Acer* spp., of *Fraxinus excelsior* and of *Tilia* spp. | Slovenia CLU55 | <0.01 | 0.01 | 0.01 | 0.98 |
| FT10 | Thermophilous *Fagus sylvatica* forests | Slovenia CLU55 | 0.06 | 0.01 | 0.01 | 0.98 |
| FT11 | Forests and woodlands of thermophilous broadleaves | Slovenia CLU56 | 0.08 | <0.01 | 0.02 | 0.98 |
| FT12 | Forest of *Pinus sylvestirs* and of *Pinus nigra* | Slovenia CLU55 | 0.02 | <0.01 | 0.02 | 0.98 |
| FT13 | Forests of *Abies alba* and of *Picea abies* on carbonate and mixed bedrock | Slovenia CLU54 | 0.01 | <0.01 | 0.10 | 0.90 |
| FT14 | Forests of *Abies alba* and of *Picea abies* on silicate bedrock | Slovenia CLU54 | 0.04 | <0.01 | 0.06 | 0.94 |
| FT15 | Forests of *Larix decidua* and Woodlands of *Pinus mugo* | Slovenia CLU54 | 0.01 | 0 | 0 | 1.00 |

Forest types and model classifiers were then linked to the appropriate climate unit of the archive index database (Figure 1; Table 1). Thus, we provided a suitable set of ecological parameters which are necessary and affect the carbon cycle, such as the decay rate of dead organic matter, the transfer of carbon between carbon pools and litterfall characteristics. The majority of forest types were classified in the following climate units: Slovenia–CLU55 (central and southern Slovenia), Slovenia-CLU45 (Savinja and Styria regions) and Slovenia–CLU54 (Montane and altimontane forests at higher altitudes in the Alps and Pohorje).

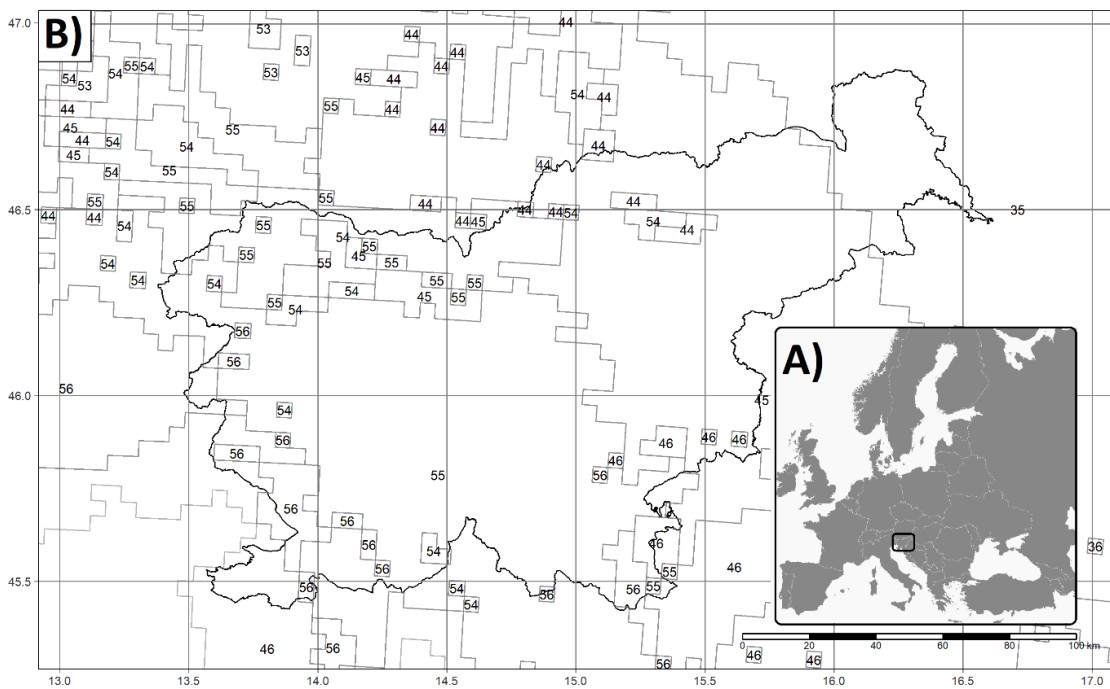

**Figure 1.** Map of (**A**) Slovenia and (**B**) Europe and climate units from the archive index database.

2.2.2. Development of Growth Curves and Determination of Age

To initialise and run the CBM-CFS3, at least the following input data are required for each classifier: areas by age classes and yield curves. Areas were calculated from the Forest Compartment Database and yield curves from the PSP database.

Since in Slovenia we do not classify forest stands into age classes when planning forest management, and moreover we manage a significant area of uneven-aged stands, we had to define the (approximate) age of each stand on PSPs, first. We did that following the ensuing methodology. The dominant diameter $DBH_{dom}$ per plot (i.e., mean DBH of the 100 thickest trees per hectare) was calculated for the two most recent measurements ($DBH_{dom.0}$ and $DBH_{dom.1}$); both were then classified into DBH classes of 5 cm. For each PSP, the average diameter increment of dominant trees ($I_{DBH.dom}$) was calculated as the difference between $DBH_{dom.1}$ and $DBH_{dom.0}$. Based on the $I_{DBH.dom}$, the transition periods that dominant trees needed to overgrow the observed DBH class were calculated for each forest type. By summing these transition periods from that of the lowest DBH class to that of the observed DBH class on a plot, we estimated the age of a stand on each PSP. When the age of a stand was calculated, we assumed that dominant trees need 20 years to achieve the DBH measurement threshold of 10 cm, regardless of the forest type they grow in. With all that data, we could finally classify stands on PSPs into age classes spanning 20 years each. We classified 10 age classes, each ranging 20 years, but the oldest age class (AGEID09) included all stands with an age greater than 181 years (Table 2).

**Table 2.** Share of harvesting for each age class.

| Age Class | Age | Share |
|---|---|---|
| AGEID00 | 0–20 | 0.000 |
| AGEID01 | 21–40 | 0.001 |
| AGEID02 | 41–60 | 0.001 |
| AGEID03 | 61–80 | 0.019 |
| AGEID04 | 81–100 | 0.233 |
| AGEID05 | 101–120 | 0.431 |
| AGEID06 | 121–140 | 0.230 |
| AGEID07 | 141–160 | 0.070 |
| AGEID08 | 161–180 | 0.010 |
| AGEID09 | 181+ | 0.006 |

The forest area was calculated for each forest type by summing the areas of the compartments of that forest type. The total area of each forest type was then proportionally divided into mixture types and age classes according to the share of PSPs of each combination of mixture type and age class. Since PSPs are generally not tallied in young forests, the youngest age class (0–20 years) included the areas of young forests obtained from the stand map of the SFS.

For each classifier, the yield curve was calculated using Equation (1), where GS represents growing stock, AGE is the middle age of an age class, and a, b, c represent model parameters. For all forest types, yield curves for 43 classifiers were developed (Figure 2) according to the method described above.

$$GS = a + b \times AGE + c \times AGE^2 \tag{1}$$

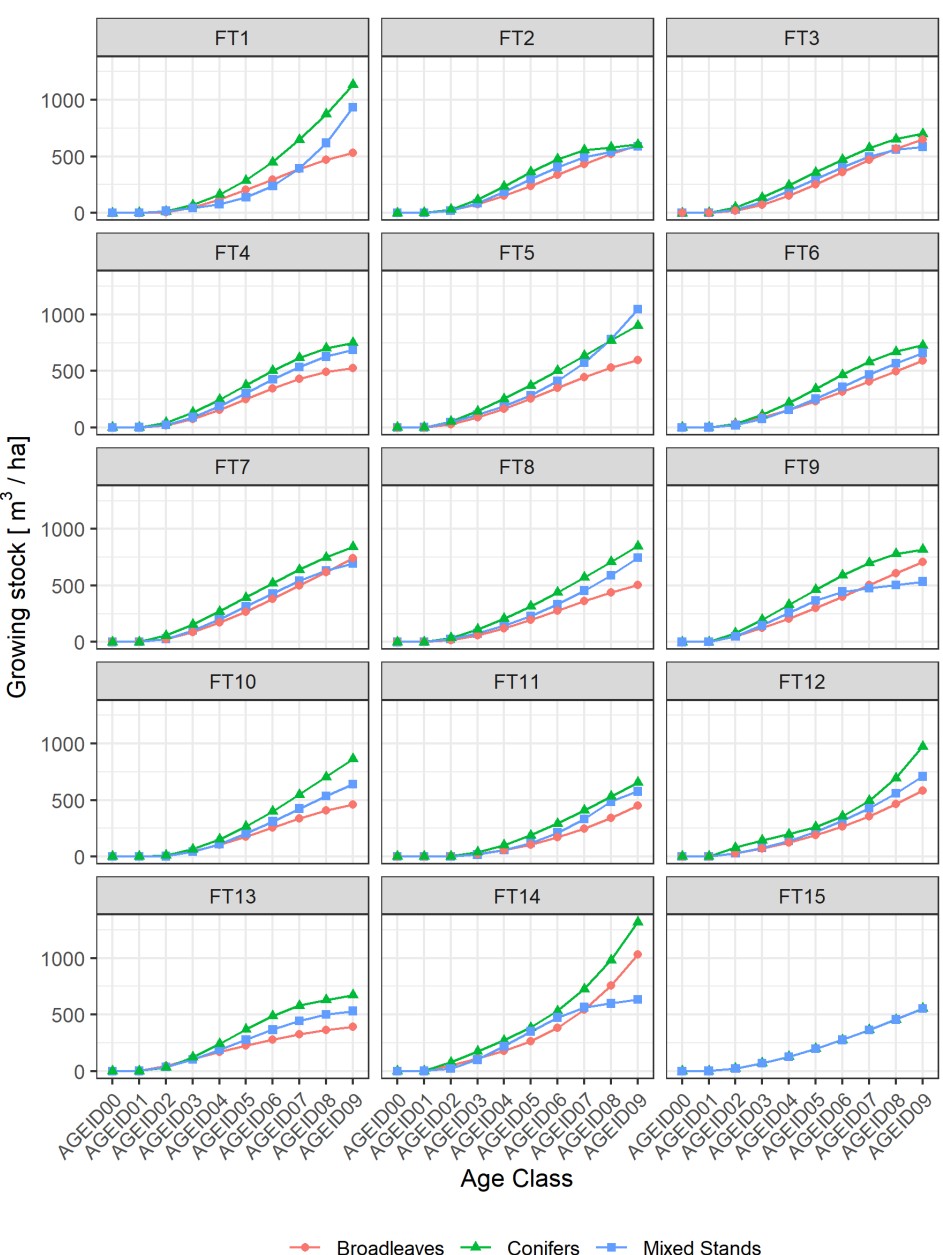

**Figure 2.** Growth curves by forest type (FT) and tree species mixture categories. Forest type abbreviations are defined in Table 1.

### 2.3. Harvesting Scenarios

The time-related carbon sink dynamics in Slovenian forests was analysed according to five hypothetical forest management scenarios (Figure 3): (1) business-as-usual (BAU), (2) harvesting in line with current forest management plans (PLAN), (3) more frequent natural hazards (HAZ), (4) high harvest (HH) and (5) low harvest (LH). All scenarios are based exclusively on the assumption of future harvest intensity and amount of harvested biomass, since the latter most significantly determines carbon stock dynamics.

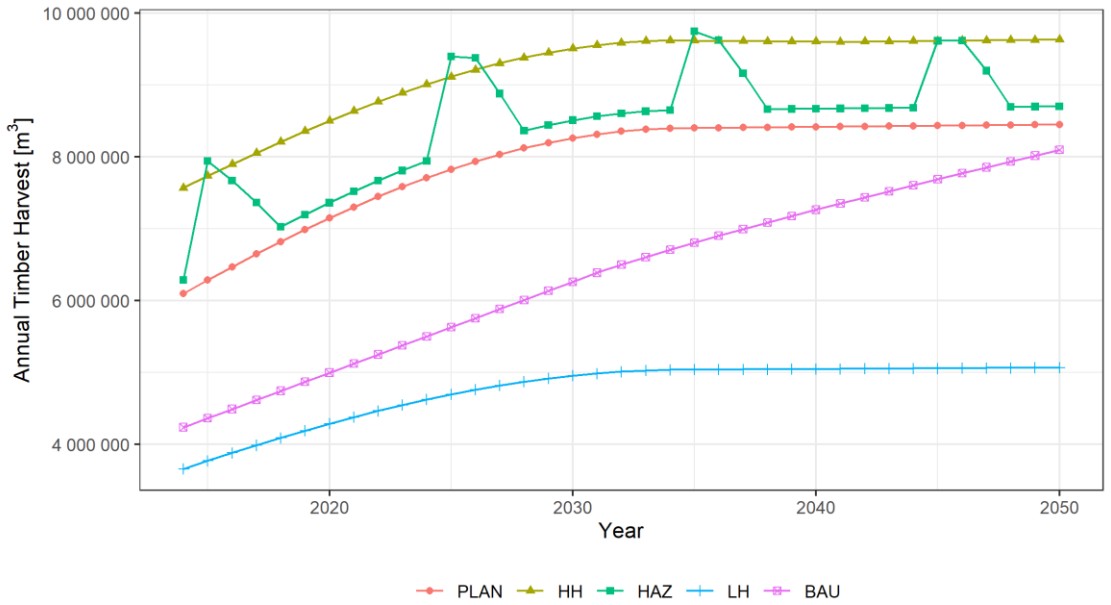

**Figure 3.** Annual timber harvest in the period 2014–2050 for five scenarios: business-as-usual (BAU), forest management planning (PLAN), natural hazards (HAZ), high harvest (HH) and low harvest (LH).

The BAU scenario is based on the realised timber harvest in the past 10 years. Importantly, the BAU scenario does not account for additional harvesting due to natural disturbances. Based on the amount of timber harvested in the period 2004–2019 [9] and omitting data from years in which natural disturbance events occurred (2014–2016, 2018), the harvest intensity was regressed as a function of time ($R^2$ = 0.789). Prior to 2030, we used the calculated function to estimate annual timber harvest, while it was assumed that the harvest amount would increase by 1% per year in the period 2030–2050. Thus, we assumed a progressive increase in harvesting, which is expected to exceed 8 million m$^3$ in 2050. When determining the PLAN scenario, we followed the recommendations from forest management plans where the annual allowable cut is defined at a rate of 75–90% of the annual stand volume increment [9]. The annual harvested amount in the PLAN scenario was determined by fitting the negative exponential function, with the dependent variable being the increase in the rate of planned harvest compared to the planned harvest in the previous year ($R^2$ = 0.358). To calculate the adjusted values, we used the planned harvest in 2004 of 4,162,662 m$^3$ [35] as a starting point. Until 2035, the degree of increase of the harvested biomass took place pursuant to the calculated function; after 2035, a constant level of increase of 0.04% per year was used. The HAZ scenario was also based on the PLAN scenario and assumed a 3% increase in harvesting levels on an annual basis, as well as four extraordinary natural disturbances, appearing in an interval of approximately 10 years and assuming an increase in harvest intensity over the following three years, which is usually a consequence of bark beetle outbreaks [12]. The HH and LH scenarios are primarily based on the assumption of the annual allowable cut of the PLAN scenario, but the intensity of harvesting in the HH scenario was increased by approximately 30% and in LH reduced by 40%. With the selected increase and reduction, we wanted to obtain a

higher range of annual harvesting levels, which could be potentially beneficial for the evaluation of harvesting levels on carbon stocks.

*2.4. Distribution of Felling by Forest Types and Age Classes*

In a disturbance matrix, we defined the type and amount of felling by classifiers and age classes for each year of the simulation. The annual felling rate was first allocated to classifiers in line with the existing volume shares in 2014. The highest harvesting amounts were, therefore, directed towards classifiers with the largest volume share in the inventory database, such as FT4 and FT5 mixed forests (Table 1). For each classifier, the harvest amount was then distributed by age classes according to the proportions representing the actual timber harvest in the last 10 years (Table 2). These shares were estimated on the basis of information from the SFS harvesting databases. Two disturbance types, namely commercial thinning and final felling were defined, whereby for each year a portion of final felling was allocated to the oldest forests. The stand maps were used to determine the percentages of final felling, which were 16% (BAU, PLAN and LH), 18% (HAZ) and 20% (HH). Final felling affects the regeneration of stands, since the model starts a new succession stage with age equal to 0.

*2.5. Evaluation of the Model Simulations*

The development of carbon stock dynamics in Slovenian forests was simulated using the CBM–CFS3 for all five scenarios from 2014 to 2050. Carbon stocks were compared with the official estimates reported by Slovenia in the National Inventory Report (NIR) [36] in the framework of the UNFCCC for the period 2014–2018. The mean absolute percentage error (MAPE) [37] between the simulated and reported values of above- and below-ground biomass was calculated. In addition, we estimated and compared the net annual carbon stock change for the carbon pools. Conversion of carbon (C), expressed in tonnes, to carbon dioxide $CO_2$ (tonnes) was performed by use of Equation (2), where the constant 44/12 is used to convert carbon density into units of $CO_2$. Negative values represent removals from the atmosphere, while positive values represent emissions to the atmosphere.

$$CO_2 = C \times (44/12) \tag{2}$$

**3. Results**

The accumulation of carbon stock in living above- and below-ground biomass coincides with the predicted harvest intensity for each scenario, whereby the accumulation is the highest for the LH and BAU scenarios. Both of these scenarios envisage the lowest harvesting up to 2050 compared to the other scenarios (Figure 4). An increase in carbon stocks in above-ground biomass is generally expected, as the anticipated harvesting levels are mainly lower than the annual volume increment in Slovenian forests of approximately 9 million $m^3$ [9]. After 2025, the HH scenario assumes that harvesting will exceed the total volume increment (Figure 5). By 2050, the carbon stock in above-ground biomass increases by 28.4% (LH), 19% (BAU), 10% (PLAN), 6.5% (HAZ) and 1.2% (HH) compared to the base year of 2014. In the case of below-ground biomass, these shares were the highest in the LH (16.9%), BAU (8.4%) and PLAN (1.1%) scenarios, while stocks in below-ground biomass in the HAZ and HH scenarios decreased by 1.2% and 6.3%, respectively. The analysis of the temporal dynamics, taking into account the change in carbon stocks in living biomass (i.e., above- and below-ground), show that Slovenian forests will remain a carbon sink under all scenarios, with the exception of the HH and HAZ scenarios in some years (Figure 5). The average annual $CO_2$ sinks in the period 2014–2050 according to the scenarios are as follows: −4187 (LH), −2455 (BAU), −1093 (PLAN), −648 (HAZ) and −5 (HH) Gg $CO_2$.

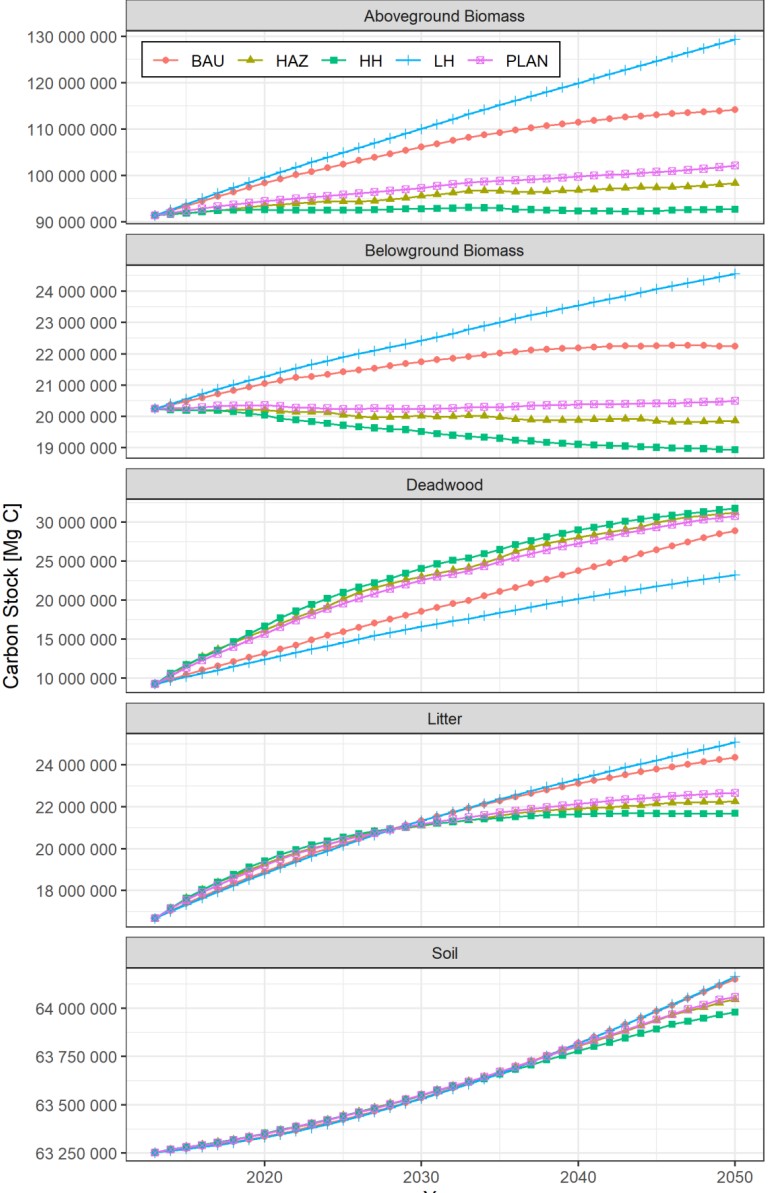

**Figure 4.** Carbon stocks in megagrams (Mg) for five forest carbon pools according to harvesting scenarios.

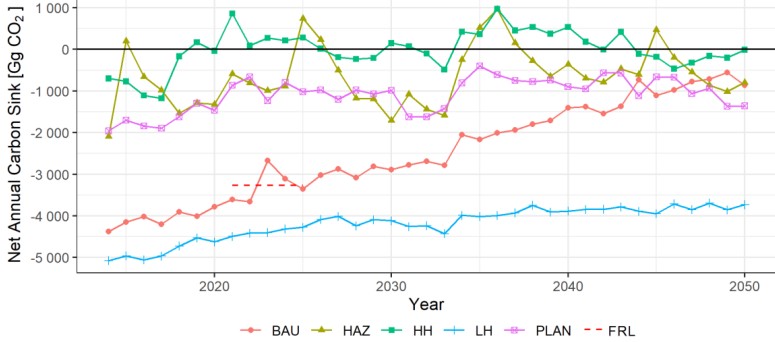

**Figure 5.** Carbon sink scenarios for Slovenian forests up to 2050 in gigagrams (Gg) of $CO_2$. Note that only living biomass pools (above- and below-ground) are compared here. The short dashed line depicts the Forest Reference Level (FRL), which refers to the proposed forest reference level determined by Slovenia for the 2021–2025 period [38].

Higher harvesting levels have a negative effect on carbon stock in litter and soil, but also a positive effect on the accumulation of carbon stock in deadwood (Figure 4). The effect on the carbon stock in the soil and litter becomes more pronounced after 2030 and 2035, respectively, when the differences between the scenarios begin to increase. Carbon stock in deadwood has doubled (scenario LH) or tripled (scenarios HH, PLAN and HAZ) in 2050 compared to the base year 2014. However, deadwood accumulation largely depends on conservation measures and the attitude of forest owners towards deadwood, and thus the results might not reflect the true dynamics in Slovenian forests.

The impact of harvesting on carbon stocks in stands of different age classes (Figure 6) indicates the ageing of forests for all scenarios, which depends on the allocation of harvested biomass to specific age classes and a relatively low share of final fellings. Taking into account the mean values of the age classes, the average age of forests in 2014 is 87 years, and in 2050 it is between 110 (HH) and 119 (LH) years.

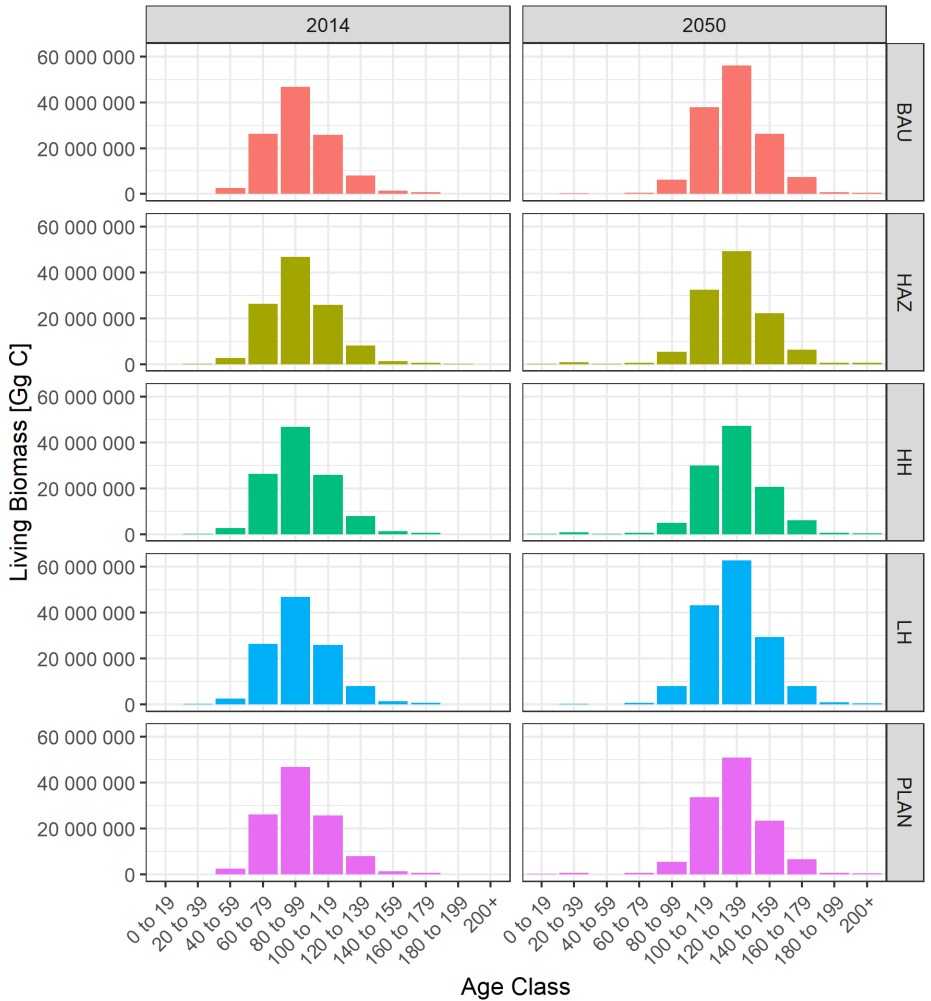

**Figure 6.** Comparison of carbon stocks in living biomass in gigagrams (Gg) distributed by age classes between 2014 and 2050 according to the harvesting scenarios. Living biomass includes above- and below-ground biomass.

The results of the CBM-CFS3 simulations were compared with the official estimates reported by the Slovenian NIR in 2020 for the period 2014–2018 [36] in the framework of the UNFCCC (Table 3). The results of model simulations show satisfactory match with the officially reported estimates of carbon stocks, where the MAPE is greater for below-ground biomass. MAPE was on average 4.5% for above-ground biomass and 9.16% for below-ground biomass. Better results were obtained for the BAU and LH scenarios, while for the HAZ, HH and PLAN scenarios, which assumed higher harvest

intensity, the MAPE were greater. More specifically, MAPE for the comparative period 2014–2018 were lowest in the LH scenario (2.8% for above- and 7.8% for below-ground biomass) and highest for the HH scenario (5.7% for above- and 10.1% for below-ground biomass).

**Table 3.** Comparison between simulated and reported (United Nations Framework Convention on Climate Change (UNFCCC)) estimates of carbon stocks in living biomass in megagrams for Slovenian forests in the period 2014–2018.

| Pool | Year | BAU | HAZ | HH | LH | PLAN | UNFCCC |
|---|---|---|---|---|---|---|---|
| Aboveground Biomass | 2014 | 92,449,926 | 91,937,056 | 91,620,158 | 92,609,249 | 91,904,714 | 95,569,258 |
| | 2015 | 93,472,363 | 91,947,659 | 91,842,930 | 93,812,247 | 92,367,470 | 96,646,110 |
| | 2016 | 94,460,483 | 92,135,505 | 92,136,303 | 95,028,069 | 92,837,497 | 97,694,381 |
| | 2017 | 95,475,138 | 92,412,609 | 92,465,503 | 96,225,439 | 93,311,079 | 98,824,962 |
| | 2018 | 96,436,244 | 92,809,421 | 92,543,887 | 97,376,402 | 93,743,549 | 99,857,742 |
| Belowground Biomass | 2014 | 20374144 | 20,264,093 | 20,201,345 | 20,405,736 | 20,260,448 | 21,945,152 |
| | 2015 | 20,484,398 | 20,198,478 | 20,188,414 | 20,555,456 | 20,261,042 | 22,203,899 |
| | 2016 | 20,591,113 | 20,189,400 | 20,197,102 | 20,718,664 | 20,293,635 | 22,456,301 |
| | 2017 | 20,722,173 | 20,179,104 | 20,187,943 | 20,874,974 | 20,337,153 | 22,727,850 |
| | 2018 | 20,826,925 | 20,201,283 | 20,154,696 | 21,013,010 | 20,345,674 | 22,977,130 |

## 4. Discussion

### 4.1. The Impact of Harvesting on Carbon Stock Dynamics in Slovenian Forests

The CBM-CFS3 has been recognized as one of the most reliable tools for simulating carbon stock dynamics in forest ecosystems [39] and has often been used in similar simulations in Europe [29,40] and globally [27,41]. Initially, this model was developed for even-aged forests which are commonly found in far northern geographic regions. However, the model can also be applied to mixed and uneven-aged forests if the yield curves objectively describe the development of forest biomass over time [29].

The carbon stock dynamics in above- and below-ground biomass simulated by the CBM-CFS3 show a satisfactory match with the values reported by Slovenia in the framework of the UNFCCC (Table 3), whereby the results for above-ground biomass yielded lower MAPE. Greater MAPE in terms of below-ground biomass are to be expected since the entire calibration of the model is based on the inventory database which only encompasses above-ground biomass. National reports in the framework of the UNFCCC are based on the National Forest Inventory (NFI) [42–44], which is a different data source than that used in our study. Our simulations are based on data of the SFS which is collected in the framework of forest inventories for the purpose of forest management planning. The disadvantage of this data is that only 1/10 of forest management compartments are updated every year. Therefore, the data used are not completely up to date. Furthermore, the NFI database is based on individual tree species, which are systematically measured every 5–6 years and as such represent a more consistent data source. Further reasons for general underestimation of reported values could be related to the parameterization of the CBM-CFS3 model. Also, the selected comparison period (2014–2018) might be unsuitable for comparison due to many natural disturbances during this period. Discrepancies between the observed and simulated carbon stocks from the CBM-CFS3 model were also reported for red spruce (*Picea rubens*) forest in Eastern Canada [45].

The results of our simulations realistically reflect the dependence between the assumed future harvest and projected net carbon sinks. All simulated scenarios, with the exception of HH and occasionally HAZ, indicate the possibility of significantly higher annual harvesting than that seen in the previous two decades, which would still guarantee a carbon sink on a national scale. Higher harvest intensity would also make sense in terms of ensuring the sustainability of all forest functions. However, in the first accounting period from 2021 to 2025, the Forest Reference Level (FRL) will need to be considered when planning the amount of harvesting in all EU member states [46]. The

proposal of a draft delegated act assumes an FRL value of $-3270.2$ Gg $CO_2$ eq per year for Slovenia in the period 2021–2025 [38], which according to our simulations, means limiting the harvest to about 6 million $m^3$ per year. The exact threshold at which the FRL value will be exceeded is hard to specify, since net sinks largely depend on the tree species being harvested [47]. In addition, the overall carbon balance also depends on wood use, and hence the inclusion of harvested wood products (HWP) and substitution effects in simulations of carbon pool developments can have a substantial impact on the overall carbon balance [48,49]. Finally, to meet the commitment under the LULUCF regulation [7], Slovenia should enhance the carbon sink in forests, promote the use of wood as a material and energy source, and ensure the stability of the forest land area.

The intensity and structure of harvesting are two important components when changing the age structure of forests. In the base year of 2014, the largest share of forests is represented in the age class of 81–100 years, while the largest share of forests in the final year of 2050 will be in the class of 101–120 years. The latter is due to the distribution of the harvest into age classes, where the largest share is directed towards semi-mature and old forests (approximately 90% of the harvested forests are in the age classes 80–140 years). Due to the short simulation period (37 years), all classifiers were advanced by only one age class. Nevertheless, we can conclude from our simulations that forest rejuvenation is a long-term process in which most of the harvest should be directed to the oldest forests and the regeneration of these forests, which would ensure a balanced age structure in the coming decades. Regardless of the accounting rules after 2030, the harvest in Slovenia will have to be increased towards the middle of this century to ensure that forests act as a carbon sink and remain resilient over the long term.

*4.2. Sources of Uncertainty*

The CBM-CFS3 has proven to be a reliable tool for simulating carbon stock dynamics at the national level. Yield curves represent key input data for estimating carbon stock dynamics with the CBM-CFS3 model. In the even-aged forests of northern latitudes, volume yield curves are often well-known [50]. In our case, these were calculated on the basis of the PSP data on growing stock and their estimated age from the calculated transition periods. The disadvantage of this procedure is that the yield curves thus calculated already contain a part of the harvest, which is reflected in lower growing stock. We estimated that the curves include an average of 2.5 million $m^3$ of harvest per year, which coincides with the low harvesting in the period 2000–2009 [9]. The adequacy of the correction choice was confirmed by simulations in the preliminary phase, in which the harvesting and volume increment were equalised and the carbon sinks were approximately 0. For uneven-aged heterogeneous managed forests, which are typical for Slovenia, yield curves are difficult to calculate precisely. We selected a relatively simple regression equation, while more complex approaches would include additional explanatory variables, e.g., site index, elevation and slope [43]. However, based on the satisfactory match between simulated and reported estimates of carbon stocks for the overlapping period 2014–2018, we assume the developed yield curves realistically reflect the development of stand biomass over time.

Sources of uncertainty are also related to the choice of ecological parameters for tree species mixture categories. In our study, we defined three different categories of species mixture (see Section 2.2.1), which must be connected to existing tree species in the archive index database. The broadleaf forests (BRD) were connected with the category OB_SI (other broadleaf trees of Slovenia) and coniferous forests (CON) with the category OC_SI (other coniferous trees of Slovenia). The mixed forest category (MIX) is not available for Slovenia, so we connected mixed forests with the category OB_SI. We also checked other options (e.g., OC_SI), but matching with reported estimates resulted in larger MAPE. This choice seems reasonable, as broadleaves are more common in Slovenia's mixed forests than conifers.

The results of our simulations provide a realistic picture of the impact of harvesting on net carbon sinks projections up to 2050 for Slovenian forests. However, some other studies reported a distinct turnaround of carbon sequestration after 2050 [51,52], which could also happen in our simulations, if a

longer period was considered. In addition to harvesting, changes in carbon stocks in living biomass may also be affected by other processes, such as the degree of tree mortality and changes in forest areas, which were not covered in our study and should be addressed in future studies.

## 5. Conclusions

In this study, we used the well-established and tested CBM-CFS3 model to quantify the impact of various harvesting scenarios on the forest carbon dynamics of Slovenian forests. The results of the simulations showed a satisfactory match with officially reported estimates under the UNFCCC and highlighted the importance of properly selected harvesting intensities to ensure carbon sinks in future decades. Harvesting below 9 million m$^3$ per year will ensure that Slovenian forest land is a carbon sink, while harvesting should be around 6 million m$^3$ per year to comply with the FRL.

**Author Contributions:** J.J.: writing—original draft preparation, methodology (data preparation and model run), formal analysis and visualization. M.K. writing—review and editing, methodology (data preparation and yield curve development) and discussion. B.M.: writing—review and editing, project administration, conceptualization and funding acquisition. All authors have read and agreed to the published version of the manuscript.

**Funding:** The research was financially supported by the project "LIFE ClimatePath 2050-Slovenian Path Towards the Mid-Century Climate Target" (LIFE16 GIC/SI/000043) and the target research project "A Review and Evaluation of Forest Development Models for Forest Management Planning at Different Spatial Scales" (V4-1821), financed by the Ministry of Agriculture, Forestry and Food and the Slovenian Research Agency. The latter also finances the Programme Groups "Forest Biology, Ecology, and Technology" (P4–0107) and "Forest, Forestry, and Renewable Forest Resources" (P4–0059), in which the authors work.

**Acknowledgments:** We thank two anonymous reviewers, who provided many useful suggestions which are now included in the manuscript.

**Conflicts of Interest:** The authors declare no conflict of interest. The funders had no role in the design of the study; in the collection, analyses, or interpretation of data; in the writing of the manuscript, or in the decision to publish the results.

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
