# Peer review of "The Effect of Harvesting on National Forest Carbon Sinks up to 2050 Simulated by the CBM-CFS3 Model: A Case Study from Slovenia"

_forests, doi:10.3390/f11101090_

Round 1

Reviewer 1 Report

The manuscript is well-done and I believe is ready for publication with a few minor edits. The authors have done a very good job putting together the input variables, executing the model, and providing some very relevant scientific information for carbon accounting and policy decisions. The use of the CBM model is well-supported by similar work outside the region of development (Canada) and appears to have been modified sufficiently to represent the current are of interest (Slovenia) adequately. I have a few comments for the authors to consider and one major item that should be addressed before the paper is published.

The most glaring omission as far as I can tell is the lack of reporting of the model components other than aboveground and belowground biomass. The authors describe the five forest carbon pools (L84-85), but deadwood, mineral soil, and litter have no further mention in the manuscript. It is not clear what impact these pools have on the total carbon budget, and to achieve a net ecosystem carbon balance (NECB) estimate, these pool changes need to be included. I imagine that there could be losses in the soil pools due to the increased decomposition and use of soil nutrients for growing forest stands. While these losses might be small, they could alter the overall estimates. If the authors have purposely left these estimates out of the study, this should be explicitly stated in the approach. Also, it would be important to note if this is appropriate for the policy guidelines that are mentioned in the introduction (L45-49).

Specific comments:

L 25       Is ‘reported’ the correct word? Perhaps ‘calculated’ is more accurate?

L36-37  Are the percentages derived from citation #3? If so, I suggest moving the citation to the end of the sentence. If not, please provide the citation.

L73        Add ‘the’ in front of Northern.

L97-99  The climate information to adjust decay rates is good, but there is no information on decay rates, or organic matter turnover provided in the paper. Litter and organic carbon turnover and loss are an important loss vector for total forest carbon accounting.

L140      Is ‘social class’ defined somewhere else? Please be more specific regarding this variable.

Table 1 and Figure 1       I suggest combining the data from Figure 1 into table 1. The figure doesn’t add a lot of compelling information and the summary lends itself better to a table.

L178      I work in an area with much older forests, so this number surprised me. I suggest adding some background on the history of forest management that perhaps has harvested all the timberland over the past two centuries? Some perspective on stand ages and management would be helpful. Perhaps this is just the managed forest and the older, late seral stands are not included in the analysis?

L179-186            This section is confusing to me. I believe that this method is used to estimate the stand age and then place the stand into an age class? The goal of this methodology should be more clearly and explicitly stated as a topic sentence. The age classes are the goal, but the method is also to estimate stand age.

L196      The variables for this equation need to be defined. Age seems obvious, but what age exactly (age class?) should be defined.

Figure 4. Different symbols along with the colored lines would help define the data better. This would also facilitate a greyscale figure and assist people who are color-challenged.

L234      Why were 30% and 40% chosen for HH and LH respectively?

L257      I suggest adding using Mg (Megagrams) in addition or instead of ‘tonnes’. Many scientists use Mg to report forest carbon stocks and change.

L261      Define the variables in this equation.

Figure 6. As in Figure 4, use different symbols. Define kt in the y-axis label in the figure caption. Also, in the caption, note that the FRL is the small dashed line in the figure.

Figure 8.             I suggest converting the data to a table and including columns with the percent change for each scenario.

L385-386            What is ‘mio’?

Author Response

Reviewer 1

The manuscript is well-done and I believe is ready for publication with a few minor edits. The authors have done a very good job putting together the input variables, executing the model, and providing some very relevant scientific information for carbon accounting and policy decisions. The use of the CBM model is well-supported by similar work outside the region of development (Canada) and appears to have been modified sufficiently to represent the current are of interest (Slovenia) adequately. I have a few comments for the authors to consider and one major item that should be addressed before the paper is published.

The most glaring omission as far as I can tell is the lack of reporting of the model components other than aboveground and belowground biomass. The authors describe the five forest carbon pools (L84-85), but deadwood, mineral soil, and litter have no further mention in the manuscript. It is not clear what impact these pools have on the total carbon budget, and to achieve a net ecosystem carbon balance (NECB) estimate, these pool changes need to be included. I imagine that there could be losses in the soil pools due to the increased decomposition and use of soil nutrients for growing forest stands. While these losses might be small, they could alter the overall estimates. If the authors have purposely left these estimates out of the study, this should be explicitly stated in the approach. Also, it would be important to note if this is appropriate for the policy guidelines that are mentioned in the introduction (L45-49).

Authors: For the initial submission, we did not include other carbon pools, i.e. deadwood, litter and soil, since we do not have reliable data sources to verify the simulations. However, based on your suggestion, we have now added also the remaining pools, but we still remain focused on the above- and below-ground biomass.

Specific comments:

L 25 Is ‘reported’ the correct word? Perhaps ‘calculated’ is more accurate?

Authors: We have changed the phrase ‘reported’ and now use ‘calculated’.  

L36-37 Are the percentages derived from citation #3? If so, I suggest moving the citation to the end of the sentence. If not, please provide the citation.

Authors: Yes, the references are derived from citation [3], which is now moved to the end of the sentence.

L73 Add ‘the’ in front of Northern.

Authors: Corrected.

L97-99 The climate information to adjust decay rates is good, but there is no information on decay rates, or organic matter turnover provided in the paper. Litter and organic carbon turnover and loss are an important loss vector for total forest carbon accounting.

Authors: We have now also included the results for litter, deadwood and soil organic carbon.

L140      Is ‘social class’ defined somewhere else? Please be more specific regarding this variable.

Authors: Here we describe the Slovenian permanent sampling plots in general, where (Kraft’s) social class is one of measured/estimated variable, but it is not used in the analysis later. Therefore, we have decided to remove this information not to confuse readers.  

Table 1 and Figure 1 I suggest combining the data from Figure 1 into table 1. The figure doesn’t add a lot of compelling information and the summary lends itself better to a table.

Authors: We have now joined information from (previous) Table 1 and (previous) Figure 1 into new Table 1.

L178 I work in an area with much older forests, so this number surprised me. I suggest adding some background on the history of forest management that perhaps has harvested all the timberland over the past two centuries? Some perspective on stand ages and management would be helpful. Perhaps this is just the managed forest and the older, late seral stands are not included in the analysis?

Authors: We changed the text accordingly to reviewer’s remark. In Central Europe production cycles in managed even-aged forests are 80-140 y, sometimes up to 180 y long. In Slovenia, they are mainly 100-140 y long, in some forest types up to 170-180 y. Thus, we used age classes up to 180 years, but in the last age class (AGEID09), all the remaining stands were included with age greater than 180 years. We have now described this more precisely in the manuscript.

L179-186 This section is confusing to me. I believe that this method is used to estimate the stand age and then place the stand into an age class? The goal of this methodology should be more clearly and explicitly stated as a topic sentence. The age classes are the goal, but the method is also to estimate stand age.

Authors: We supplemented and partly changed the text trying to more clearly express the goal of this calculation which was to estimate the age of a stand on a PSP and more clearly and precisely describe the method applied. For explanation, in Slovenia we do not operate with stand age when planning and managing forests. We relate silvicultural measures and harvest to developmental stages or stand type (namely regeneration, pole stage, mature stage, rejuvenation stage, uneven-aged stand), for which we can roughly estimate the age through dominant stand DBH, the average diameter increment of dominant trees and based on that calculated transition periods (i.e. years needed for dominant trees to overgrow from one 5-cm DBH class to another). These transition periods are then summed for DBH classes from the first (i.e. the lowest dominant DBH class) to the one to which dominant DBH of a stand was classified.

L196 The variables for this equation need to be defined. Age seems obvious, but what age exactly (age class?) should be defined.

Authors: AGE was the middle age of each age class. We supplemented the text.

Figure 4. Different symbols along with the colored lines would help define the data better. This would also facilitate a greyscale figure and assist people who are color-challenged.

Authors: As suggested, different symbols are now added to each scenario. (The same was done for all other figures).

L234      Why were 30% and 40% chosen for HH and LH respectively?

Authors: Both scenarios are based on the PLAN scenario, and with the selected increase/decrease, we wanted to obtain high range of selected harvesting levels. With HH, we wanted to exceed the annual volume increase, while with LH, we wanted to keep the harvesting levels low, similar to the period 1991 – 2013. We have added one sentence to inform users about our selection.

L257      I suggest adding using Mg (Megagrams) in addition or instead of ‘tonnes’. Many scientists use Mg to report forest carbon stocks and change.

Authors: We agree with your suggestion and now use megagrams in all figures and inside the text. One exception is Figure 5, where we use Gg to keep the same units as used in the FRL draft proposal.

L261      Define the variables in this equation.

Authors: Variables from Equation 2 are already defined inside the text: “Conversion of carbon (C), expressed in tonnes, to carbon dioxide CO2 (megatons) was performed by using Equation 2”. There are no other variables in this equation.

Figure 6. As in Figure 4, use different symbols. Define kt in the y-axis label in the figure caption. Also, in the caption, note that the FRL is the small dashed line in the figure.

Authors: Done as suggested.

Figure 8. I suggest converting the data to a table and including columns with the percent change for each scenario.

Authors: We have converted (previous) Figure 8 into new Table 3. The percentage change for each scenario is still given only within text, since we would like to avoid duplicated results in tables and text.

L385-386            What is ‘mio’?

Authors: We apologize for this typo. We now use the full expression, i.e. million.

Reviewer 2 Report

Reviewer Recommendation and Comments for Manuscript forests- 950125:

The effect of harvesting on national forest carbon sinks up to 2050 simulated by the CBM-CFS3 model: a case study from Slovenia

The article applies the Carbon Budget Model of the Canadian Forest Sector (CBM-CFS3) for simulating future carbon stocks in Slovenian forests under five different harvesting scenarios over a period of 36 years. The topic is important; the analysis, and the procedures involved, are described in a very transparent and reproducible manner, and in almost perfect English. A publication is therefore desirable. Some of the results, however, would need more discussion in my mind; the main issue here is that the simulated estimates of carbon stocks do not seem to fit too well to the reported ones (cf. Fig.8). I therefore suggest accepting the article after some (mostly minor) changes.

Specific comments

  • Line 47-49: You may want to add a cross reference to the respective EU regulation here [VO (EU) 2018/841] (EU, 2018)
  • L55-56: What is the reason for the relatively low harvests 1991-2013? Please add a short hint here. (If the reason is related to an uneven age structure, then the insights generated from the relatively short simulation period cannot be extended to the future, which would be relevant for result interpretation – see also later comment to line 344).
  • L62: This remains unclear to me. Depending on the given age and stand structures, (a) ageing needs not necessarily be associated with reduced resistance to natural disturbances; moreover, (b) older forest stands usually store more timber, and hence the economic value of the forests increases rather than decreases. Possibly you mean something different here? (I.e., @a: the accumulation of carbon implies that the risk of carbon release due to a natural disturbance becomes higher; @b: after a culmination, growth rates [but not the growing stock, or its economic value] decrease)?
  • L78, “long-term”: In a forest growth context, a 36 year simulation period is definitively not “long” (cf. line 344!). I’d rather say here: “In this study, we present simulations of the forest carbon sink with the CBM-CFS3 for the time period 2014-2050. The main objective of the study was to evaluate the effect of harvesting impacts in Slovenia on forest carbon storage according to five harvesting scenarios”.
  • L98, “customised”: I’m not sure whether “adapted” or “parametrised” might fit better here.
  • 146: it remains unclear what a “prevalent forest community” is, and why & by whom it is “proposed”; suggestion: delete (=> “Forest type was determined according to the typology of Slovenian forest sites, see tab.1”)
  • 165/Figure 1: Colours are undistinguishable in this (and in the other) figures, e.g. when printing the article in b/w, or for readers suffering from colour blindness. Suggestion: use grey/black/white in fig.1; solid/dotted/dashed lines in fig.3; different symbols (rather than a point throughout) in fig.4 and fig.6; a combination of more contrasting colours and dotting in fig.5; omit different colours in fig.7; apply different shadings in fig.8 (and, as far as possible, try using the same colours/shadings/dotting throughout the different graphs).
  • 192-198: (Just as a remark – modelling the (remaining) growing stock directly from age with a simple quadratic model, and applying this as a yield function, reduces complex relations between tree age, site quality, total volume growth, and human harvesting decisions into one quite simple formula. While this is not “wrong” in any respect, it may be a reason for parametrisation problems, which I suppose might be a reason for the apparent misfit between observed and simulated carbon stocks in fig.8. Therefore, it would be desirable if the authors could provide some additional information about model fit for the graphs which are depicted in Fig.3).
  • 255, “The relative error between the simulated and reported values ... was calculated”: It would be interesting to present these results in the article.
  • 261 (formula 2): This is not quite correct for a conversion from tonnes to tonnes; the conversion formula is simply CO2 = C*(44/12) (i.e. drop the minus sign and the 10^-3 term)
  • 269, “harvesting will exceed the total volume increment, resulting in net emissions”: This is a misinterpretation, since harvested wood is usually not oxidised immediately; rather, this depends on the kind of wood utilisation (see e.g. Schulze et al. (2020); Bösch et al. (2019) demonstrate that including HWP and substitution effects in simulations of carbon pool developments can have a substantial impact on the overall carbon balance). Hence, I suggest deleting “resulting in net emissions” here, and adding a short explanation in the discussion section explaining that “harvesting” is not identical to “emitting carbon” (or equivalently, that the focus on the forest sector suggested by the LULUCF reporting rules may be misinterpreted if HWP and substitution are not being considered in an overall carbon balance).
  • 275-276, “Slovenian forests will remain a carbon sink under all scenarios, with the exception of the HH scenario”: This description does not seem to fit well to fig.6. According to fig.6, both HH and HAZ show “emissions” in some of the years, but not in all years. Even on average, the text reports in line 277-278 that all scenarios (including HH) lead to negative emissions over the time period considered.
  • 287-291/fig.6: For clarity, Y-axis and figure heading should read “net annual carbon stock changes” rather than “carbon sink”
  • 298, “results of the model simulations match well with the officially reported estimates of carbon stocks”: This seems not to fit well to the data shown in fig.8, where reported carbon stocks consistently are much higher than any of the simulated ones; even the variation between the various simulated stocks is less than the variation between simulated and reported stocks. (As an addition, it would be desirable if the authors could provide not only the development of carbon stocks but also of the net changes, for a comparison of simulated and reported data).
  • 317, “show a consistent match”: Again, I do not fully agree here; in my view figure 8 shows a consistent underestimation of carbon stocks in each of the simulations. It might be worthwhile to discuss the possible reasons here into some more detail (e.g.: the parametrisation of the model might suffer from general deficits; there might be a problem in the transition from observed to projected data [which would probably affect the transition only, but not the relations between the results of the different simulations – this might be visible when additionally showing the development of the net changes, as suggested above]; the comparison period 2014-2018 might be unsuited for a comparison due to many natural disturbances in this period [according to lines 216-217, almost every year in this period has been affected by natural disturbances] etc.).
  • 321-322: This specific reason (different data sources) is not very convincing in my mind, specifically since the authors only mention that there are differences, but do not explain these differences in detail. Moreover, it is argued in lines 325-326 that “this data realistically reflect[s] the standing volume and areas of various forest types”. If this is the case, i.e. if the SFS sample is representative, then differences between SFS and NFI cannot be hold responsible for the abovementioned underestimation.
  • 330-332: This is but a speculation, since the article has not analysed “the sustainability of all forest functions”.
  • 344, “short simulation period”: Agree! It might make much sense to add here that long-term results may look very differently (as an example, long term simulations for a country with a quite uneven forest age structure show a distinct turnaround of C sequestration after 2050 (Schweinle et al., 2017); cf. also (Knauf et al., 2016).
  • 350, “remain resilient…”: although I fully agree, this is not a result of the simulations.
  • 362, “yield curves are difficult to calculate precisely”: Would it make sense to use curves provided by single tree stand simulators, like the one developed by Pretzsch et al. (2002) for Germany?
  • 363, 383, “good match”: see comments above.
  • As part of the discussion, it might also make sense to compare results to those of other (similar) simulation excercises in other countries.

Lit. quoted

  • Bösch, M., Elsasser, P., Rock, J., Weimar, H., Dieter, M., 2019. Extent and costs of forest-based climate change mitigation in Germany: accounting for substitution. Carbon Management 10, 127-134.
  • EU, 2018. Regulation (EU) 2018/841 of the European parliament and of the council of 30 May 2018 on the inclusion of greenhouse gas emissions and removals from land use, land use change and forestry in the 2030 climate and energy framework, and amending Regulation (EU) No 525/2013 and Decision No 529/2013/EU. Amtsblatt der Europäischen Union L156, 1-25.
  • Knauf, M., Joosten, R., Frühwald, A., 2016. Assessing fossil fuel substitution through wood use based on long-term simulations. Carbon Management 7, 67-77.
  • Pretzsch, H., Biber, P., Dursky, J., 2002. The single tree-based stand simulator SILVA: construction, application and evaluation. Forest Ecology and Management 162, 3-21.
  • Schulze, E.D., Sierra, C.A., Egenolf, V., Woerdehoff, R., Irslinger, R., Baldamus, C., Stupak, I., Spellmann, H., 2020. The climate change mitigation effect of bioenergy from sustainably managed forests in Central Europe. GCB Bioenergy 12, 186-197.
  • Schweinle, J., Köthke, M., Englert, H., Dieter, M., 2017. Simulation of forest-based carbon balances for Germany: a contribution to the ‘carbon debt’ debate. Wiley Interdisciplinary Reviews: Energy and Environment, e260.

Author Response

Reviewer 2

The article applies the Carbon Budget Model of the Canadian Forest Sector (CBM-CFS3) for simulating future carbon stocks in Slovenian forests under five different harvesting scenarios over a period of 36 years. The topic is important; the analysis, and the procedures involved, are described in a very transparent and reproducible manner, and in almost perfect English. A publication is therefore desirable. Some of the results, however, would need more discussion in my mind; the main issue here is that the simulated estimates of carbon stocks do not seem to fit too well to the reported ones (cf. Fig.8). I therefore suggest accepting the article after some (mostly minor) changes.

Specific comments

Line 47-49: You may want to add a cross reference to the respective EU regulation here [VO (EU) 2018/841] (EU, 2018)

Authors: The suggested reference is added.

L55-56: What is the reason for the relatively low harvests 1991-2013? Please add a short hint here. (If the reason is related to an uneven age structure, then the insights generated from the relatively short simulation period cannot be extended to the future, which would be relevant for result interpretation – see also later comment to line 344).

Authors: There are several reasons for low harvest in the period 1991-2000: after the independence of Slovenia, we adopted new Forest Act and started with denationalization. Due to rising living standards, there was lower economic dependence of forest owners on their forest. There were also socioeconomic changes in Slovenian population, where younger people moved closer to urban places and were less interested in farming and working with forests. In those times, forest policy emphasized accumulation of timber stock aiming at allowable cut to be up to 60 % of annual increment, nature conservation has become strongly established, decline of wood processing industry which also had an inhibitory effect on the felling and use of wood. After the year 2000, and especially after 2007, felling is gradually increasing. In 2007, a new national forest programme was adopted which strategically set the direction for greater allowable cut (i.e. up to 75 % of annual increment). Thus, low harvest in this period can be partly attributed to forest management planning and partly to low interest of forest owners. We have now briefly mentioned the key factors in the manuscript and also provided a reference, where those factors are discussed more deeply.

L62: This remains unclear to me. Depending on the given age and stand structures, (a) ageing needs not necessarily be associated with reduced resistance to natural disturbances; moreover, (b) older forest stands usually store more timber, and hence the economic value of the forests increases rather than decreases. Possibly you mean something different here? (I.e., @a: the accumulation of carbon implies that the risk of carbon release due to a natural disturbance becomes higher; @b: after a culmination, growth rates [but not the growing stock, or its economic value] decrease)?

Authors: We partly agree with the reviewer’s remark, but not completely. The main disturbance agent in Slovenian forests is wind. Wind usually damage older even-aged stands (also older cohorts within group selection forests which are frequent in Slovenia) and previously damaged stands, but much less younger stands like pole stage stands (e.g. Klopčič et al., 2009). Also bark beetles are important agent, which also damage older trees and older stands (e.g. Klopčič et al., 2009; Hlasny et al., 2019). Thus aging of forests increase the probability of a stand to be damaged.

In addition, aging of forests quite often decrease the economic value of forests. In Slovenia, the main tree species are European beech (Fagus sylvatica) and Norway spruce (Picea abies). Timber quality of the first is highly related to the age of a tree (Poljanec and Kadunc, 2017), since the probability of red heart increases sharply at older trees of larger dbh (Knoke, 2003). If beech tree has inside of a trunk a red heart its economic value is (much) lower. Norway spruce, though, doesn’t have such a characteristics which would decrease its economic value while aging, but if damaged by wind or bark beetles (or other disturbance agents) its economic value is much lower.

Considering both issues we can conclude that ageing of forests reduces their resistance to natural disturbances (i.e. wind, bark beetles attacks) and their economic value.

We shortly supplemented the text. However, there are many papers, which report various positive effects of ageing on forest, so we also briefly reference those.

L78, “long-term”: In a forest growth context, a 36 year simulation period is definitively not “long” (cf. line 344!). I’d rather say here: “In this study, we present simulations of the forest carbon sink with the CBM-CFS3 for the time period 2014-2050. The main objective of the study was to evaluate the effect of harvesting impacts in Slovenia on forest carbon storage according to five harvesting scenarios”.

Authors: We agree with your suggestion and replaced the sentence with yours.

L98, “customised”: I’m not sure whether “adapted” or “parametrised” might fit better here.

Authors: Authors of the paper, which we reference here, also used the expression ”customised” (https://ec.europa.eu/jrc/en/scientific-tool/eu-archive-index-database-customised-carbon-budget-model-cbm-cfs3). We believe it is the best to use the same wording as in original publication.

146: it remains unclear what a “prevalent forest community” is, and why & by whom it is “proposed”; suggestion: delete (=> “Forest type was determined according to the typology of Slovenian forest sites, see tab.1”)

Authors: All forests in Slovenia are divided into approximately 53,000 forest compartments. For each compartment, we have information related to species mixture, DBH, yield, forest community, etc... However, in each compartment, different forest communities might exist, but one is prevailing (for each community, the area is also given). Forest communities in Slovenian forests are mapped and proposed by local forest engineers/district foresters in the scope of forest planning. They visit each compartment once per ten years and describe the forest, including forest communities. But as you suggested, not to confuse readers, we have now modified the sentence based on your suggestions. 

165/Figure 1: Colours are undistinguishable in this (and in the other) figures, e.g. when printing the article in b/w, or for readers suffering from colour blindness. Suggestion: use grey/black/white in fig.1; solid/dotted/dashed lines in fig.3; different symbols (rather than a point throughout) in fig.4 and fig.6; a combination of more contrasting colours and dotting in fig.5; omit different colours in fig.7; apply different shadings in fig.8 (and, as far as possible, try using the same colours/shadings/dotting throughout the different graphs).

Authors: Thank you very much for this suggestion. All figures are now modified in a way, that can be easily understood in b/w format. 

192-198: (Just as a remark – modelling the (remaining) growing stock directly from age with a simple quadratic model, and applying this as a yield function, reduces complex relations between tree age, site quality, total volume growth, and human harvesting decisions into one quite simple formula. While this is not “wrong” in any respect, it may be a reason for parametrisation problems, which I suppose might be a reason for the apparent misfit between observed and simulated carbon stocks in fig.8. Therefore, it would be desirable if the authors could provide some additional information about model fit for the graphs which are depicted in Fig.3).

Authors: For running the CBM model, we used the data provided for simulating forest development using the EFISCEN model (Nabuurs et al. 2007; Verkerk et al. 2011). This model uses similarly simple equation to predict volume growth based on stand age (Schelhaas et al., 2002; Schelhaas et al., 2007). Since it is simple and efficient, we transferred it to our case and try to express growing stock with as simple equation as possible. We are aware that it does not depict all influencing factors, but equations are forest type (site) specific thus they partly depict differences between sites. For example, site quality is partly considered by developing yield curves for each forest type, but we are aware that site quality varies also within the same forest type. Total volume growth, which was also exposed by the reviewer, is closely related to site quality, while harvesting is actually included into the yield curves which is discussed in the section 4.2. We agree with the reviewer that this might be a reason for misfit between observed and simulated carbon stocks, but we think it is not the only nor the greatest reason.

In the Discussion, we now also discuss additional options for the development of growth curves.

255, “The relative error between the simulated and reported values ... was calculated”: It would be interesting to present these results in the article.

Authors: Thank you for this warning. We now use the mean absolute percentage error (MAPE), and the results are presented in the last paragraph of the Results section.

261 (formula 2): This is not quite correct for a conversion from tonnes to tonnes; the conversion formula is simply CO2 = C*(44/12) (i.e. drop the minus sign and the 10^-3 term)

Authors: Indeed, formula did not convert tonnes to tonnes. So we have now corrected the formula based on your comment.

269, “harvesting will exceed the total volume increment, resulting in net emissions”: This is a misinterpretation, since harvested wood is usually not oxidised immediately; rather, this depends on the kind of wood utilisation (see e.g. Schulze et al. (2020); Bösch et al. (2019) demonstrate that including HWP and substitution effects in simulations of carbon pool developments can have a substantial impact on the overall carbon balance). Hence, I suggest deleting “resulting in net emissions” here, and adding a short explanation in the discussion section explaining that “harvesting” is not identical to “emitting carbon” (or equivalently, that the focus on the forest sector suggested by the LULUCF reporting rules may be misinterpreted if HWP and substitution are not being considered in an overall carbon balance).

Authors: This is indeed very valuable suggestion. We have deleted the suggested sentence and now added a few lines to the Discussion related to HWP and their effect on the overall balance. We have also added the suggested references.

275-276, “Slovenian forests will remain a carbon sink under all scenarios, with the exception of the HH scenario”: This description does not seem to fit well to fig.6. According to fig.6, both HH and HAZ show “emissions” in some of the years, but not in all years. Even on average, the text reports in line 277-278 that all scenarios (including HH) lead to negative emissions over the time period considered.

Authors: Yes, you are correct and we now more precisely describe the model simulation results: “The analysis of the temporal dynamics, taking into account the change in carbon stocks in living biomass (i.e. above- and below-ground), show that Slovenian forests will remain a carbon sink under all scenarios, with the exception of the HH and HAZ scenarios in some years (Figure 5).”

287-291/fig.6: For clarity, Y-axis and figure heading should read “net annual carbon stock changes” rather than “carbon sink”

Authors: We have now modified the y axis heading as suggested. In addition, kt (kilotons) was changed into Gg (gigagrams) to be consistent with other figures and text.

298, “results of the model simulations match well with the officially reported estimates of carbon stocks”: This seems not to fit well to the data shown in fig.8, where reported carbon stocks consistently are much higher than any of the simulated ones; even the variation between the various simulated stocks is less than the variation between simulated and reported stocks. (As an addition, it would be desirable if the authors could provide not only the development of carbon stocks but also of the net changes, for a comparison of simulated and reported data).

Authors: We agree that the results could also be better and have now changed our wording from “good match” into “satisfactory match”.

317, “show a consistent match”: Again, I do not fully agree here; in my view figure 8 shows a consistent underestimation of carbon stocks in each of the simulations. It might be worthwhile to discuss the possible reasons here into some more detail (e.g.: the parametrisation of the model might suffer from general deficits; there might be a problem in the transition from observed to projected data [which would probably affect the transition only, but not the relations between the results of the different simulations – this might be visible when additionally showing the development of the net changes, as suggested above]; the comparison period 2014-2018 might be unsuited for a comparison due to many natural disturbances in this period [according to lines 216-217, almost every year in this period has been affected by natural disturbances] etc.).

Authors: Well, in some cases, MAPE was relatively good, i.e. 2.7 % for LH scenario, but we do agree that there is definitely a room for improvement. We have changed our wording and now describe our results as satisfactory. We have also added a few sentences where we discuss possible reasons for the consistent underestimation.

321-322: This specific reason (different data sources) is not very convincing in my mind, specifically since the authors only mention that there are differences, but do not explain these differences in detail. Moreover, it is argued in lines 325-326 that “this data realistically reflect[s] the standing volume and areas of various forest types”. If this is the case, i.e. if the SFS sample is representative, then differences between SFS and NFI cannot be hold responsible for the abovementioned underestimation.

Authors: We have added additional explanation regarding the differences between the two databases.

330-332: This is but a speculation, since the article has not analysed “the sustainability of all forest functions”.

Authors: We agree with the comment, the statement about the representativeness was deleted. We have added additional explanation regarding the differences between the two databases.

344, “short simulation period”: Agree! It might make much sense to add here that long-term results may look very differently (as an example, long term simulations for a country with a quite uneven forest age structure show a distinct turnaround of C sequestration after 2050 (Schweinle et al., 2017); cf. also (Knauf et al., 2016).

Authors: This is interesting argument. We have now also mentioned this possibility and referenced the suggested literature in the last paragraph od the Section 4.2.

350, “remain resilient…”: although I fully agree, this is not a result of the simulations.

Authors: We agree with you, but this is a discussion, where we try to put our results into a more practical context and therefore argue that this sentence does not harm here.

362, “yield curves are difficult to calculate precisely”: Would it make sense to use curves provided by single tree stand simulators, like the one developed by Pretzsch et al. (2002) for Germany?

Authors: We definitely agree that there are other options, which might yield better yield curves. But we worked with forest types, which are mostly mixed; and as far as I know, SILVA provides yield curves for individual species. We have indicated possible improvements in the manuscript now.

363, 383, “good match”: see comments above.

Authors: We have now changed our wording and use “satisfactory match” in the manuscript.

As part of the discussion, it might also make sense to compare results to those of other (similar) simulation excercises in other countries.

Authors: We have now extended the number of references and related our results to similar studies.

Lit. quoted

  • Bösch, M., Elsasser, P., Rock, J., Weimar, H., Dieter, M., 2019. Extent and costs of forest-based climate change mitigation in Germany: accounting for substitution. Carbon Management 10, 127-134.
  • EU, 2018. Regulation (EU) 2018/841 of the European parliament and of the council of 30 May 2018 on the inclusion of greenhouse gas emissions and removals from land use, land use change and forestry in the 2030 climate and energy framework, and amending Regulation (EU) No 525/2013 and Decision No 529/2013/EU. Amtsblatt der Europäischen Union L156, 1-25.
  • Knauf, M., Joosten, R., Frühwald, A., 2016. Assessing fossil fuel substitution through wood use based on long-term simulations. Carbon Management 7, 67-77.
  • Pretzsch, H., Biber, P., Dursky, J., 2002. The single tree-based stand simulator SILVA: construction, application and evaluation. Forest Ecology and Management 162, 3-21.
  • Schulze, E.D., Sierra, C.A., Egenolf, V., Woerdehoff, R., Irslinger, R., Baldamus, C., Stupak, I., Spellmann, H., 2020. The climate change mitigation effect of bioenergy from sustainably managed forests in Central Europe. GCB Bioenergy 12, 186-197.
  • Schweinle, J., Köthke, M., Englert, H., Dieter, M., 2017. Simulation of forest-based carbon balances for Germany: a contribution to the ‘carbon debt’ debate. Wiley Interdisciplinary Reviews: Energy and Environment, e260.

References used by authors in the Response Letter

Hlásny, T., Krokene, P., Liebhold, A., Montagné-Huck, C., Müller, J., Qin, H., Raffa, K., Schelhaas, M.-J., Seidl, R., Svoboda, M., Viiri, H. (2019) Living with bark beetles: impacts, outlook and management options. From Science to Policy 8. European Forest Institute.

Klopčič, M., Poljanec, A., Gartner, A., Bončina, A. (2009) Factors related to natural disturbances in mountain Norway spruce (Picea abies) forests in the Julian Alps. Ecoscience, 16 (1), 48-57.

Knoke, T. (2003) Predicting red heartwood formation in beech trees (Fagus sylvatica L.). Ecological Modelling 169: 295–312.

Nabuurs, G.J., Pussinen, A., van Bursselen, J., Verkerk, P.J. (2007) Future harvesting pressure on European forests. European Journal of Forest Research 126: 391–400.

Poljanec, A., Kadunc, A. (2013) Quality and timber value of European beech (Fagus sylvatica L.) trees in the Karavanke region. Croatian Journal of Forest Engineering 34: 151-165.

Schelhaas, M.-J., Nabuurs, G.J., Sonntag, M., Pussinen, A. (2002) Adding natural disturbances to a large-scale forest scenario model and a case study for Switzerland. Forest Ecology and Management, 167: 13-26.

Schelhaas, M.-J., Eggers, J., Lindner, M., Nabuurs, G.J., Päivinen, R., Schuck, A., Verkerk, P.J., Werf, D.C. van der, Zudin, S. (2007). Model documentation for the European Forest Information Scenario model (EFISCEN 3.1.3). Alterra report 1559 and EFI technical report 26. Alterra and European Forest Institute, Wageningen and Joensuu, p. 118.

Verkerk, P.J., Antilla, P., Eggers, J., Lindner, M., Asikainen, A. (2011) The realisable potential supply of woody biomass from forests in the European Union. Forest Ecology and Management 261: 2007–2015.
